# Variation of Gene Expression in the Endemic Dinaric Karst Cave-Dwelling Bivalve Mollusk *Congeria kusceri* during the Summer Season

**Annalisa Scapolatiello** [1,2], **Chiara Manfrin** [1], **Samuele Greco** [1], **Tomislav Rončević** [3], **Alberto Pallavicini** [1,4], **Sanja Puljas** [3] **and Marco Gerdol** [1,*]

1  Department of Life Sciences, University of Trieste, 34127 Trieste, Italy
2  Department of Biology, University of Padua, 35122 Padova, Italy
3  Faculty of Science, University of Split, 21000 Split, Croatia
4  Anton Dohrn Zoological Station, 80121 Naples, Italy
*  Correspondence: mgerdol@units.it

**Abstract:** The cave systems of the Neretva River basin in the Dinaric Karst are home to *Congeria kusceri*, one of the very few known examples of stygobiotic bivalve mollusks, which displays several unique life history traits and adaptations that allowed its adaptation to the subterranean environment. This endemic species is undergoing rapid decline, most likely linked with habitat degradation, which might seriously threaten its survival in the next few decades. Unfortunately, the urgent need for effective conservation efforts is hampered by the lack of effective regulations aimed at preserving remnant populations as well as by our limited knowledge of the biology of this species. Although the precise factors underlying the disappearance of *C. kusceri* from its type locations are not entirely clear, the alteration of seasonal changes in water temperatures and alkalinity is most likely involved, as these are the main drivers of shell growth, spawning, and the onset and progression of the new gametogenic cycle. While these aspects have so far only been studied using morphological and physiological observations, future conservation efforts would certainly benefit from an integrated multidisciplinary approach. Here, using RNA-sequencing, we provide an overview of the modulation of gene expression recorded in five key tissues (the mantle, gonads, gills, adductor muscle, and digestive gland) between early June and late September. We highlight the presence of markedly tissue-specific responses, with the most dramatic changes affecting the gonads, mantle, and gills. We further identified a drastic switch in the use of energy budgets between the two periods, with evidence of ongoing shell growth and high metabolic activity in the mantle and gills at the early time point, followed by a massive redirection of all available energy to the gonads for the generation of new gamete primordia in early autumn.

**Keywords:** transcriptome; RNA-sequencing; expression level; Karst; subterranean; Bivalvia

## 1. Introduction

Despite their isolation and extreme conditions, subterranean freshwater environments are inhabited by several metazoans that display unique adaptations whose biological significance is comparable to those observed in other extreme aquatic environments, such as deep-sea hydrothermal vents and Antarctic waters [1,2]. Among these, the most important and widespread certainly include tissue depigmentation, the reduction or total loss of light perception systems, which is often compensated by an enhancement of other sensory systems, the reduction of metabolic rates, the adoption of a k-selected reproductive strategy, and a remarkable longevity compared to their epigean relatives [3]. These adaptations can be undoubtedly attributed to the particular conditions found in cave systems, where seasonal variations in terms of temperature, humidity, and water chemistry are greatly

reduced compared to epigean environments and in which nutrient availability is scarce and highly dependent on water supply from the surface.

Although cave-dwelling animal communities show significant variability in terms of composition depending on the geographical location and characteristics of the hypogeal systems, stygofauna includes both vertebrates (bony fish and amphibians) and invertebrates. Among the latter, crustaceans are certainly the dominant taxa, both in terms of abundance and biodiversity. Nevertheless, mollusks are not uncommon, with over 350 different species of gastropods described to date [4]. Although bivalve mollusks are commonly found in both saltwater and freshwater environments, only a very limited number of species have so far been described in subterranean cave systems. While it has not yet been ascertained whether some species belonging to the family Sphaeriidae, occasionally found in caves [5,6], can actually be considered stygobionts (i.e., adapted to caves and restricted to underground waters) or stygophiles (i.e., inhabiting caves and completing their entire life cycle there, but also occurring in surface habitats), until 2022, the only known species of obligate cave-dwelling bivalves belonged to the genus *Congeria* Bole 1962. However, *Eupera troglobia*, living in the Casa da Pedra cave (northern Brazil), has been recently recognized as the first stygobiont bivalve species living on the American continent [7]. Due to the elusive nature of these animals and the inaccessibility of their living environments, other cave-dwelling bivalve species may be discovered in the future.

*Congeria* is a genus endemic to the Dinaric Karst, first discovered 60 years ago, and can be considered a Tertiary relict whose present distribution range represents a very small part of the area originally occupied by its ancestors. Indeed, bivalves belonging to the family Dreissenidae (Gray 1940) were likely once widely distributed in the Paratethys Sea. Following the evolutionary radiation events that led first to the divergence between *Congeria* and *Dreissena* (~37.4 MYA) and then between *Congeria* and *Mytilopsis* (~22.8 MYA) [8], almost all *Congeria* species faced a phase of decline, eventually becoming nearly completely extinct from surface waters toward the end of the Miocene. However, a single lineage was able to successfully adapt to the subterranean environments of the Dinaric Karst, maintaining a substantial morphological stasis and surviving as a "living fossil" until the present day [8–10]. The genus includes three recognized extant species that display a discontinuous distribution in the Dinaric Karst: *Congeria kusceri* (Bole, 1962); *Congeria mulaomerovici* (Morton and Bilandžija, 2013); and *Congeria jalzici* (Morton and Bilandžija, 2013) [8]. Among these, *C. kusceri* has certainly been subjected to the most intense scientific studies, which have evidenced, on top of the aforementioned typical adaptations to the subterranean environment, several unique life history traits, including a very long life span [11] and a unique reproductive strategy. Indeed, *Congeria* reaches sexual maturity in a much longer period of time and produces much fewer offspring compared with its closest epigean Dreissenidae relatives, breeding early-stage larvae in maternal ctenidia and protecting juvenile individuals in mantle pouches. Annual reproductive cycles are strictly regulated by the water influx from the surface, which in turn critically depends on rain precipitation, determining marked seasonal fluctuations of both water levels (higher in winter, lower in summer) and temperature (lower in winter, higher in summer) [12]. Although other physical and chemical parameters also undergo significant variations throughout the year in the subterranean environment of the Dinaric Karst, their effects (if any) and the biology of *Congeria* are still poorly understood [13].

Although *C. kusceri* is well-adapted to these variable conditions, its life history traits are predicted to expose this species to a significant risk of extinction [14]. Although subterranean environments undoubtedly play a key active role in biological cycles, they have been so far largely overlooked by conservation policies [15], and the cave systems of the Dinaric Karst certainly make no exception. In fact, anthropic activities had a significant impact on the Neretva River basin, irreparably altering seasonal water influx to the underground Karst cave systems of the region [16]. Although such modifications are the prime suspects for explaining the sharp decline of *Congeria* populations observed over the past few decades [17], other factors, such as pollution and the presence of pathogens [18], may

also contribute to the demise of this species. Consequently, while *C. kusceri* is considered a critically endangered species by national directives [19], due to our poor understanding of the specific factors threatening its survival, the possibility of preserving residual populations in the long term remains unclear.

Here, we report a comprehensive catalog of the transcripts expressed in multiple tissues of *C. kusceri*, providing an overview of the transcriptional changes that underlie the seasonal biological cycle of this species. In addition to providing novel molecular insights into the regulation of several key biological processes, this resource will support conservation efforts by providing a reference for further gene expression studies, the identification of useful molecular markers for biomonitoring, and improving our understanding of the unique adaptations of this species to the subterranean environment.

## 2. Materials and Methods

### 2.1. Sample Collection and RNA Extraction

All *C. kusceri* individuals were collected at the Jama u Predolcu (Croatia, lat 43.04729°, lon 17.65851°) (Figure 1). Sampling permits were obtained from all relevant institutions (permission number: UP/I-612-07/18-48/81; 517-07-1-1-1-18-4), and all procedures performed in studies involving animals were in accordance with the ethical standards of the institution or practice at which the studies were conducted. Sampling, carried out by experienced divers, took place on two distinct occasions: on July 2 (T1) and on September 30, 2018 (T2). The average water temperature on sampling day T1 was 16.7 °C, while the dissolved oxygen concentration was 8.6 mg/L. On collection day T2, the water temperature was 17.2 °C and the dissolved oxygen was 6.6 mg/L. Both parameters, measured using a portable multimeter (a X-matePro MX300, Mettler Toledo, Columbus, OH, USA), were in line with the average values reported at the same periods by previous studies [11,13,20]. According to water level measurements conducted as part of this study, a drop in water level of approximately 42 cm was observed at T2, compared with T1.

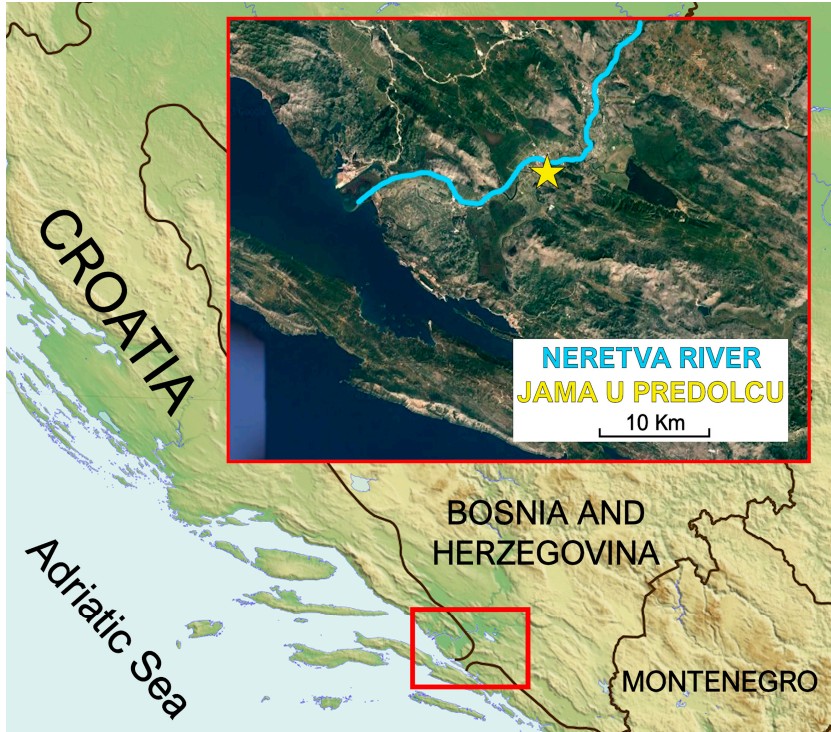

**Figure 1.** Map of the study site, showing the location of the Jama u Predolcu (marked by a five-pointed star) and the lower course of the Neretva River.

At both time points, five adult individuals of undetermined sex, ranging in shell length from 9.6 to 12.0 mm (T1) and from 9.3 to 11.7 mm (T2), were collected. Upon sacrifice, the five dissected tissues (gills, mantle, gonad, digestive gland, and adductor muscle), pooled for the five individuals, were immediately placed in RNAlater solution (Thermo Fisher, Whaltam, MA, USA) and stored at $-80$ °C until RNA extraction. Each sample was thawed, washed two times in PBS 1X, and ground with four 2.3 mm chrome-steel beads in a Bead Beater Homogenizer (BioSpec Products, Bartlesville, OK, USA) for 1'30". Total RNA was extracted with an RNA-Solv® Reagent (Omega Bio-tek Inc., Norcross, GA, USA) following the manufacturer's instructions. RNA concentration and quality were assessed using a NanoDrop 2000/2000C (Thermo Fisher) spectrophotometer.

### 2.2. Sequencing and Read Processing

Total extracted RNA was used as an input for the preparation of poly(A)-selected barcoded sequencing libraries compatible with pair-end Illumina sequencing using a TruSeq library preparation kit (Illumina, San Diego, CA, USA). RNA-sequencing was performed on an Illumina NovaSeq 6000 platform at the Genomic Core Facility of AREA Science Park (Trieste, Italy), using a 2 × 150 bp paired-end sequencing strategy. Raw reads were demultiplexed and processed with FastP, assessing their quality and removing sequencing adapters, low-quality nucleotides, and trimmed reads shorter than 75 nucleotides [21].

### 2.3. De Novo Assembly, Quality Assessment, and Annotation of the Transcriptome

Trimmed reads, down-sampled with BBNorm (https://github.com/BioInfoTools/BBMap, accessed on 1 November 2022), were used as an input to generate a de novo transcriptome assembly with Trinity (v2.8.5) [22], run with default parameters. To reduce redundancy, only the longest isoform assembled for each gene model was considered for subsequent analyses. Spurious transcripts resulting from exogenous contaminant RNAs (expected to be present due to the filter-feeding habits of *C. kusceri*) and those derived from poorly expressed and likely fragmented endogenous transcripts unlikely to carry out a biologically relevant role were detected as follows: Salmon v.1.2.1 [23] was used to calculate the coverage of all contigs (based on the combined mapping of all libraries), and all the contigs that did not reach the arbitrary threshold of five transcripts per million (TPM) [24] were removed, following the same protocol used in previous studies carried out by our team on aquatic invertebrates [25,26]. This process allowed discarding contigs that may have derived from exogenous contamination (e.g., due to the presence of food particles associated with gills and digestive glands, microbial cells associated with pathogens and/or symbionts, debris mixed with bivalve tissues as a result of filter-feeding, etc.).

All the contigs ascribable to rRNA and mitochondrial mRNA sequences were identified with BLASTn (e-value threshold = $1 \times 10^{-50}$), using available sequences from other Dreissenidae species as queries, and removed. Similarly, known abundant exogenous contaminants of viral origin [18], described in our previous paper, were discarded.

The clean transcriptome was functionally annotated with the Annotam pipeline (https://gitlab.com/54mu/annotaM, accessed on 1 November 2022), and each transcript was assigned Gene Ontology terms based on the retrieval of significant (i.e., e-value < $1 \times 10^{-5}$) DIAMOND [27] matches against the UniProtKB Swiss-Prot database [28]. The annotation of Pfam conserved protein domains was based on the identification of Pfam-A entries [29] with Hmmer v.3.1b2 [30]. The completeness of the transcriptome was assessed with BUSCO (v5.4.4) [31], analyzing the set of conserved orthologous genes of the Metazoa lineage according to OrthoDB v.10 [32].

### 2.4. Differential Gene Expression Analysis

Trimmed reads for each sequenced library were imported into the CLC Genomics Workbench v. 22 platform (Qiagen, Hilden, Germania), where differential gene expression (DGE) analysis was performed. In detail, reads were mapped against the reference transcriptome using the parameters *length fraction* = 0.75 and *similarity fraction* = 0.98. Following

a normalization by quantile of inferred read count-based gene expression levels, DGE analyses were performed for each of the five tissues with a Kal's z-test [33], comparing the expression profiles observed at T1 and T2. Differentially expressed genes (DEGs) were identified based on a False Discovery Rate (FDR) corrected *p*-value $\leq 0.05$, combined with a |Fold change| >2, i.e., selecting only the transcripts displaying an expression level more than doubled or less than halved in paired comparisons.

DEGs were subjected to gene set enrichment analyses based on hypergeometric tests [34] on GO annotations, allowing the identification of significantly enriched terms (i.e., those showing an FDR-corrected *p*-value $\leq 0.05$ and a difference between observed and expected values >3). To minimize the risk of false-positive detection, only the genes expressed in each tissue were considered as background for calculating enrichment metrics [35]. To allow a reliable comparison of gene expression levels both within and between samples, gene expression values were reported in the text using the TPM metric [24].

## 3. Results and Discussion

### 3.1. The Congeria kusceri Reference Transcriptome Assembly

Overall, following the trimming process, we generated a total of 334.4 million reads (see Table S1 for details), with an average length of 112.9 bp ($\sigma$ = 3.9), accounting for nearly 38 Gbp of sequence data. This data was assembled in 298,442 contigs, which were subsequently subjected to quality checks and decontamination as detailed in the materials and methods section. This process led to a non-redundant reference transcriptome consisting of 132,889 contigs with an average length of 736 nt and an N50 value (indicating the sequence length of the shortest contig needed to achieve 50% of the total assembly length when all contigs are sorted by length) equal to 1316 nt. Ex90N50 (a metric similar to N50 but estimated based on the subset of contigs accounting for 90% of normalized gene expression) was equal to 2036 nt (Table 1). We evaluated the completeness, fragmentation, and redundancy of the transcriptome assembly with BUSCO [31]. This analysis revealed that a high proportion of highly conserved metazoan orthologs were present and complete (93.5%), with a minimal residual level of duplication (0.8%), which might be consistent with the existence of uncollapsed divergent allelic variants, alternative splicing isoforms that could not be removed during the transcriptome refinement process, or otherwise denote the presence of species-specific gene duplications. On the other hand, a minor fraction of BUSCOs was fragmented (2.2%) or entirely missing (3.5%). Such metrics, further confirmed by the presence of over 900 contigs exceeding 5 Kb length and 45 contigs longer than 10 Kb, indicate an overall satisfying quality of the reference transcriptome assembly, which likely includes most of the transcripts expressed at biologically significant levels in *C. kusceri* during adult life under physiological conditions. While a high level of completeness was in line with our expectations due to the inclusion of the five major tissues (i.e., gills, adductor muscle, digestive gland, mantle, and gonads), we need to remark on the fact that a non-negligible fraction of the protein-coding genes encoded by the genome of this organism might not be represented in the assembly. Indeed, those exclusively expressed during early life stages, subjected to strict regulation and only expressed in response to specific stimuli, or whose expression was restricted to a very small cell population, likely did not reach sufficient read coverage to allow the assembly of the corresponding transcript. Nevertheless, since those genes were not expressed in the context of this study, they were unlikely to represent a significant obstacle to the interpretation of the data we collected.

The *C. kusceri* transcriptome was functionally annotated, allowing the association of 17,635 contigs with a GO term, 22,890 contigs with a Pfam conserved domain, and 26,941 contigs with an OrthoDB annotation. The relatively low annotation rate of the transcriptome assembly was consistent with the high number of lineage-specific gene families typically observed in bivalve genomes [36,37]. Indeed, while only 17,997 contigs found a significant BLAST hit against the UniprotKB database, which is strongly biased towards model organisms [38], a much higher number of contigs (i.e., 41,753) displayed

relevant homology with *Dreissena rostriformis*, the closest bivalve species with a fully sequenced genome available [39].

**Table 1.** Assembly and annotation metrics of the *C. kusceri* transcriptome.

| The *Congeria kusceri* Transcriptome | |
|---|---|
| number of assembled contigs | 298,442 |
| total assembly size (nt) | 97,960,735 |
| average contig size (nt) | 736.2 |
| N50 (nt) | 1316 |
| Ex90N50 (nt) | 2036 |
| complete BUSCOs (Metazoa ODB v.10) | 93.5% |
| duplicated BUSCOs (Metazoa ODB v.10) | 0.8% |
| fragmented BUSCOs (Metazoa ODB v.10) | 2.2% |
| missing BUSCOs (Metazoa ODB v.10) | 3.5% |
| contigs > 5 Kb | 920 |
| contigs > 10 Kb | 45 |
| longest assembled contig (nt) | 37,948 |
| contigs with Gene Ontology annotations | 17,635 |
| contigs with Pfam annotations | 22,890 |
| contigs with OrthoDB annotations | 26,941 |
| contigs with BLAST hits vs. UniprotKB | 17,997 |
| contigs with BLAST hits vs. *Dreissena rostriformis* | 41,753 |

Moreover, the transcriptome assembly included a high fraction of contigs lacking any detectable Open Reading Frame, which might belong to the complex and still poorly understood landscape of bivalve non-coding mRNAs [40,41]. The functional annotations of all the contigs included in the reference transcriptome assembly are provided in Table S2.

*3.2. A General Overview of Gene Expression Changes Occurring from Early July to Late September*

The transcriptional landscape of *C. kusceri* underwent significant changes over the course of the summer season, even though the magnitude of such alterations was largely tissue-dependent. With 4972 DEGs, the gills were subjected to the most significant changes (Figure 2A), followed by the gonads (3992 DEGs, Figure 2B), mantle (3252 DEGs, Figure 2C), adductor muscle (1675 DEGs, Figure 2D), and digestive gland (670 DEGs, Figure 2E). DEGs were nearly uniformly distributed between the two sampling time points, with slight variations among tissues.

Overall, 3104 out of the 10,101 DEGs identified in this study (~30.7%) were shared by more than one tissue. Significant overlaps were present between the mantle and gills, with 809 shared DEGs, most likely reflecting general physiological responses fielded by *C. kusceri* to cope with the changing environmental conditions found at the Jama u Predolcu between the two sampling periods. Nevertheless, consistent with previous reports from other bivalve species [25,42], each tissue displayed several DEGs with high tissue specificity (here arbitrarily defined as those showing expression levels >10 folds higher than other tissues) (Figure 2F). Such genes are expected to carry out extremely specialized functions that require a spatially constrained, tightly regulated expression and will therefore be analyzed in detail in the following sections due to their important functional implications. In detail, despite being the tissue showing the highest transcriptional stability, the digestive gland showed the highest level of specialization, with ~44% tissue-specific DEGs, followed by the gonads (~20%), adductor muscle (~16%), mantle (~12%), and gills (just ~5%). As a note, the identified alterations in gene expression were detected in pools of adult individuals of undetermined sex. Therefore, our experimental design did not allow us to investigate the presence of outlier individual responses, which may be to some extent linked with intraspecific differences in the genomic or epigenomic makeup of this species, or to pinpoint sex- or developmental stage-specific responses.

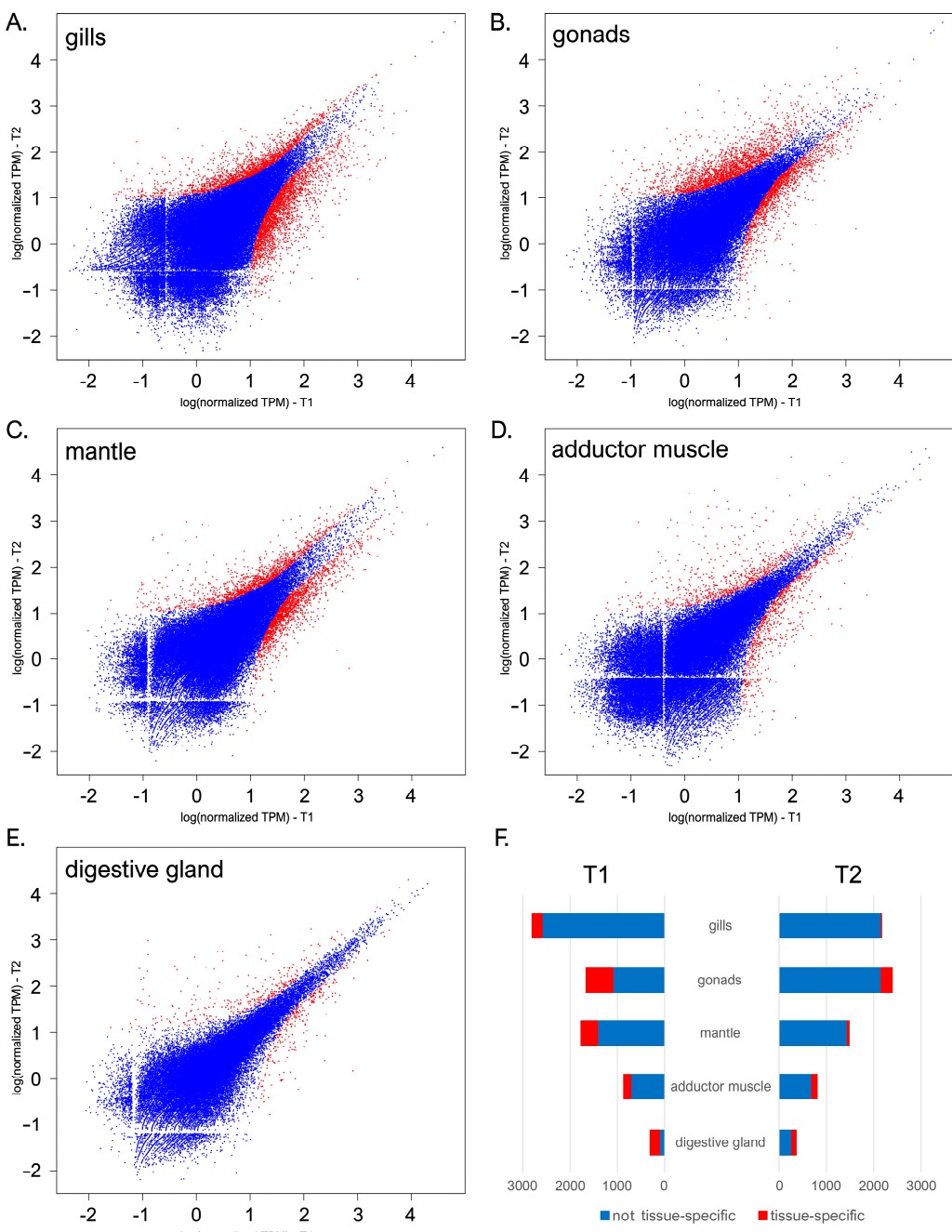

**Figure 2.** Scatter plots displaying the pairwise comparisons between the gene expression profiles observed at T1 and T2 in *C. kusceri* gills (**A**), gonads (**B**), mantle (**C**), adductor muscle (**D**), and digestive gland (**E**). Gene expression levels are reported as log (TPM + 1). Differentially expressed genes supported by a significant FDR-corrected *p*-value (i.e., <0.05) and fold change values (i.e., >2 for upregulated and <−2 for down-regulated genes) are marked with red points. Panel (**F**) reports the total number of DEGs identified in each tissue, highlighting those showing high tissue specificity.

To achieve a reliable interpretation of the trends of gene expression observed in this study, one should take into account the timing of the two sampling points with respect to the chemico-physical seasonal variations that typically characterize the Dinaric Karst subterranean system and the Jama u Predolcu in particular, whose geographical location is reported in Figure 1. Such dynamics, summarized in Figure 3, have been previously well documented by other authors and are key determinants for the progression of annual life cycles in *C. kusceri*, influencing both reproduction and growth [11,12,20]. In brief,

the combination of the seasonal atmospheric variations typical of this region and the fluctuations in water influx to the cave from the Neretva River basin usually determines a sharp change in water temperature between April and May. During the summer, the water temperature always exceeds 15 °C, reaching peaks as high as 19 °C, as opposed to the winter minima, which may even occasionally drop to about 12 °C. In parallel, water alkalinity displays similar sudden changes, sensibly decreasing during the summer season to values below 200 mg CaCO$_3$/L, compared with the higher levels usually observed during colder months [11,20]. Although variations of other chemico-physical parameters have been previously documented to occur throughout the year at the Jama u Predolcu, their effects on the biology of *C. kusceri* are still unknown [13]. Water levels also undergo significant changes, even though these variations are far less predictable and subjected to short-term fluctuations as they are strictly dependent on rainfall occurring in the area [20]. Undoubtedly, besides influencing the water temperature and levels in the Jama u Predolcu, rainfall is also responsible for providing a new influx of nutrients for *C. kusceri* through the introduction of organic matter from the surface.

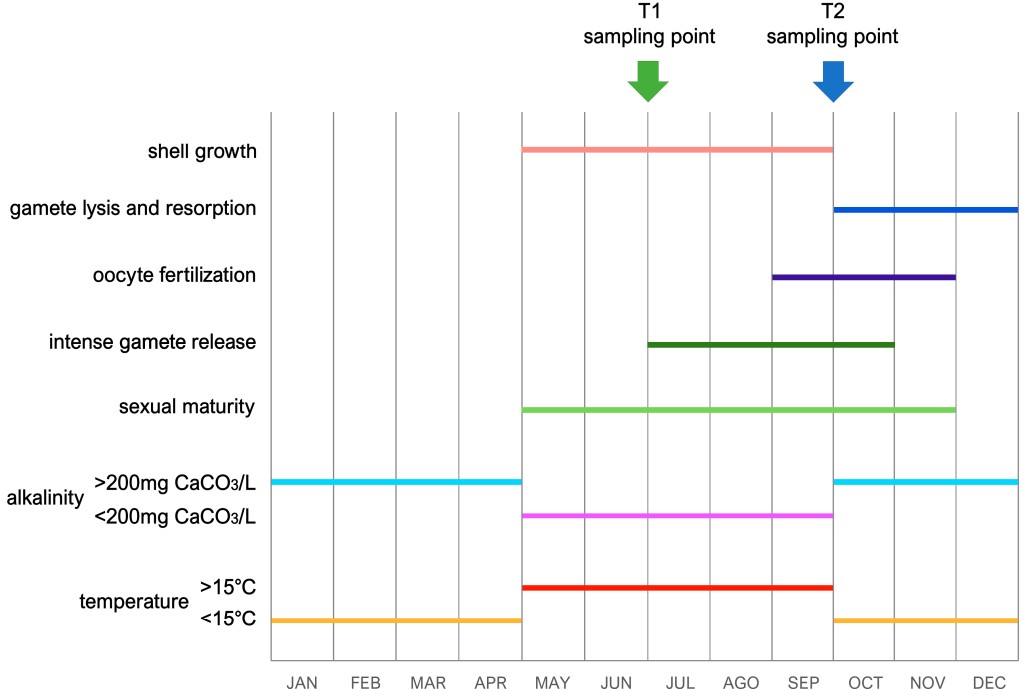

**Figure 3.** Summary of the most relevant life history traits of *C. kusceri* and of the most significant seasonal alterations of environmental parameters recorded in the Jama u Predolcu. Adapted from [11,12,37]. The graph also indicates the relationship between these factors and the timing of the two samplings (indicated by arrows).

These factors appear to be the main triggers for both shell growth and the maturation of gametes: Morton and Puljas [11] have previously observed a sudden restart of shell growth in May, coincident with the aforementioned increase in water temperature and decrease in alkalinity. Shell deposition continues up to the end of September, when both temperature and CaCO$_3$ concentration return to pre-summer levels (Figure 3). Although this direct relationship between growth and temperature mirrors the trends documented in the epigean relatives *D. polymorpha* and *Mytilopsis leucophaeata* [43,44], the observation that CaCO$_3$ uptake increases when alkalinity approaches its annual minimum would require further physiological elaboration [11]. The two sampling time points selected for this study were chosen to cover two key periods of the life cycle of *C. kusceri*, when significant physiological differences can be expected. For example, in early July (T1), the shell of *Congeria* was still in an active growth phase [11]. On the contrary, only residual shell growth can usually be observed between September and October, when T2 sampling takes

place (Figure 3). As far as the reproductive cycle is concerned, at T1, the release of gametes from the developing gonads could be observed (gametogenic acini usually start to show some empty zones in July) [12]. On the other hand, late September (T2) coincides with the sudden decrease in water temperature and increase in alkalinity, leading to intense fertilization (fertilized oocytes are usually observed in female ctenidia from September to November). This is a key period of morphological and functional changes in the gonads, which lyse and resorb unreleased gametes, starting to develop, in parallel, new gamete primordia [12] (Figure 3).

### 3.3. The Gonad Transcriptome Follows Opposing Trends of Expression Compared with Mantle and Gills

We investigated in detail the biological significance of the observed changes by performing functional enrichment analysis on the lists of DEGs obtained from the comparison between the samples collected at the two time points in each tissue. This approach allowed the identification of a similar trend of expression in the mantle and gills, which shared several enriched GO terms at both time points, in stark contrast with the gonads, which roughly displayed an opposite trend (Figure 4, Figures S1 and S2). Nevertheless, each tissue displayed its own peculiarities, with several unique annotations. The complete list of all enriched GO terms for each tissue at the two time points is provided in Tables S3–S5.

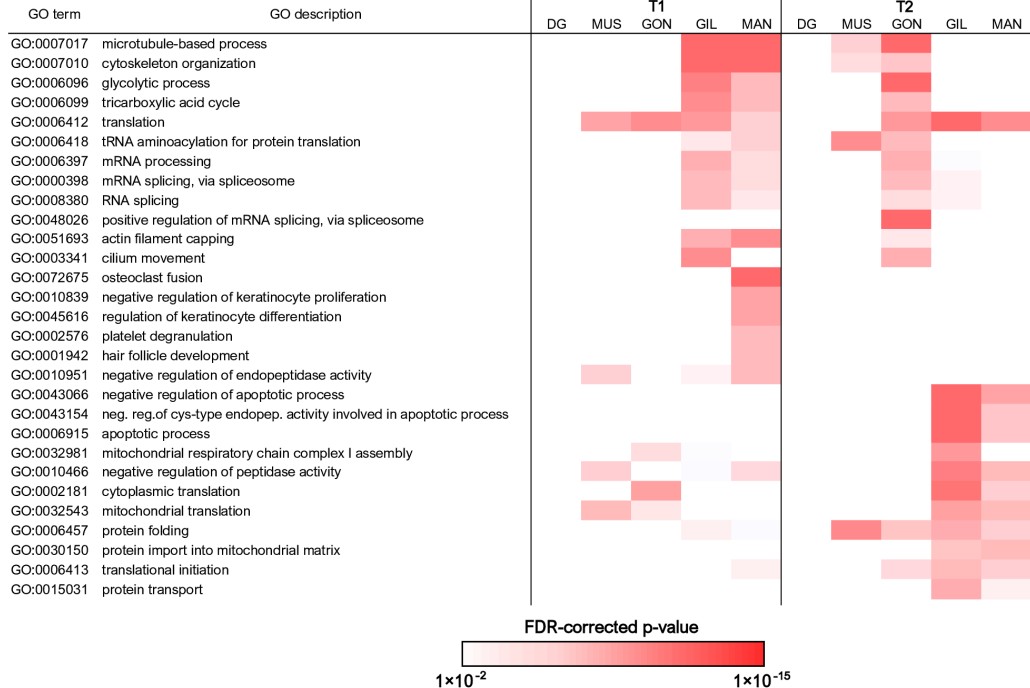

**Figure 4.** Summary of the most significantly enriched Gene Ontology biological process terms associated with differentially expressed genes. Only the terms achieving a FDR-corrected enrichment *p*-value lower than $1 \times 10^{-8}$ in at least one pairwise comparison, paired with a difference between the observed and expected number of observed DEGs associated with each GO term > 5, are shown. The complete list of all significantly enriched Gene Ontology biological process terms is reported in Table S3. DG: digestive gland; MUS: adductor muscle; GON: gonads; GIL: gills; MAN: mantle.

The functional enrichment analysis supported a higher metabolic activity at T1 in the gills and mantle, which was overall consistent with the high energetic investment expected to be placed in body growth at this time of the year, as documented at the Jama of Predolcu by previous studies [11]. Both tissues displayed a very high number of DEGs involved in mRNA maturation (and in the splicing process, in particular), indicative of high transcriptional activity. At the same time, the strong overrepresentation of terms connected with cytoskeletal dynamics, as well as those linked with glycolysis and the tricarboxylic

acid cycle, was consistent with sustained cell division and tissue proliferation. In line with this hypothesis and with available morphological observations (Figure 3), the mantle was associated with a number of additional GO terms linked with shell biomineralization.

The over-representation of the very same GO terms (excluding those linked with shell biomineralization) at T2 in gonads (Figure 4, Tables S3–S5) pointed out a marked switch in cell proliferation and associated energetic costs during the transitional phase from warmer summer months to the autumn. Considering the important structural reorganization *C. kusceri* gonads undergo at this time of the year (Figure 3), we interpret these transcriptional changes as a line of evidence supporting the regeneration of gamete primordia following intense gamete release, as discussed in detail in Section 3.5.

The similarities between the gene expression trends observed in the mantle and gills were not limited to T1. Indeed, the two tissues also shared several overrepresented GO terms at T2, even though these only occasionally underwent opposite regulation in the gonads. Such annotations were linked with mitochondrial activity and, in particular, with the mitochondrial respiratory chain (Figure 4, Tables S3–S5). Terms connected with protein translation and folding were also overrepresented, revealing a significant overlap with mitochondria-associated terms (e.g., mitochondrial translation, protein import into the mitochondrial matrix, etc.), which may suggest that mitochondrial protein synthesis was activated. Mitochondria-associated enriched GO terms were coupled with a significant over-representation of annotations linked with the negative regulation of apoptosis through the suppression of several caspases. Altogether, these findings might be linked with the important histological modifications *C. kusceri* undergoes during the summer-to-autumn transition, but possibly also to adaptation strategies specifically evolved to withstand the strong stress faced by this species during the summer season, in particular during water shortage periods [20].

Overall, consistent with the relatively low number of DEGs identified in the digestive gland (Figure 2E), this tissue was not associated with strongly altered GO biological processes, neither at T1 nor at T2 (Figure 4, Tables S3–S5). The functional enrichment of adductor muscle DEGs only revealed a few significantly enriched annotations, for the most part shared with the other tissues (Figure 4, Tables S3–S5).

While GO enrichment data can provide a general overview of the most relevant biological processes that underwent significant modulation between the two sampling time points, they do not allow an in-depth analysis of the transcriptional changes specifically occurring in each of the five tissues analyzed in this study, which will be discussed in detail in the following sections. Such in-depth tissue-specific analyses will be focused on genes displaying high tissue specificity (defined as those expressed at levels >10-fold higher than the other four tissues).

### 3.4. In-Depth Analysis of Differential Gene Expression in the Gills

Although gills displayed the highest number of DEGs among all tissues, just a very small fraction of those were tissue-specific (i.e., about 7% of all DEGs at T1 and less than 1% of all DEGs at T2, see Figure 2F). Based on transcriptomic analyses previously carried out in other bivalve species, this finding was not unexpected since gills express a broad variety of genes, likely due to the high diversity of associated cell types [25]. In fact, bivalve gills carry out multiple physiological functions, including gas exchange and filter-feeding, which are supported by a complex morphological architecture, paired with the presence of circulatory and nervous systems. Nevertheless, a few tissue-specific DEGs that showed high FC values at T1 would deserve additional attention in light of their predicted important role in the context of gill-associated functions. Namely, they encoded extracellular matrix components, such as matrilins, protocadherins, and collagen-like proteins, which may enable the establishment of the lamellar organization of branchial filaments, and a few nervous system transporters and receptors (e.g., neuronal acetylcholine receptors), which likely facilitate the coordinated movement of cilia [45].

Overall, as previously mentioned in Section 3.3, most transcriptional alterations observed in this tissue were consistent with a generalized increased rate of cell proliferation and higher metabolic activity at T1 and with an enhanced synthesis of mitochondrial protein and increased respiratory activity at T2.

*3.5. In-Depth Analysis of Differential Gene Expression in the Gonads*

As the tissue undergoing the most significant transcriptional changes in this study, with over 4000 detected DEGs (Figure 2B), the gonads were characterized by a great variety of enriched annotations (Tables S3–S5). These most likely reflect the important morphological remodeling that accompanies the *C. kusceri* reproductive cycle, with the release of gametes being closely followed by the development of new gamete primordia at the end of the summer [12] (Figure 3). As previously mentioned, the transcriptional reprogramming occurring in gonads followed an opposite direction compared with other tissues (the mantle and gills, in particular), leading to a marked activation of processes linked with energy metabolism, mRNA translation, and cytoskeletal organization at T2 (Section 3.3).

The in-depth analysis of this complex transcriptional landscape revealed an increased rate of cell division at the end of the study period, with strong evidence of enhanced DNA replication. Indeed, several DEGs were involved in nucleotide biosynthesis (e.g., ADSL, AIRC, ATIC, PFAS, and PPAT) or encoded components of the minichromosome maintenance (MCM2, MCM3, MCM4, MCM5, MCM6, MCM7, and MCMBP), condensin (NCAPG, NCAPH, SMC2 and SMC4), and cohesin (MAU2, RAD21, and STAG1) complexes. Moreover, some DEGs encode DNA polymerase subunits (POLA1, POLA2, POLD2, POLE, and POLE3) and proteins involved in post-replication DNA repair (FANCA, FANCD2, MSH2, and NBN). Other markers of ongoing mitosis encode either cell cycle regulators (e.g., AURKA, cyclin A2, MELK, MTBP, RACK1, and ZC3HC1) or proteins involved in spindle formation, microtubule polymerization, chromosome segregation, and cytokinesis (e.g., ANXA11, CLTC, HAUS3, IST1, KATNA1, KNTC1, MIS18A, MKI67, NDC80, NUP43, RPS3, SEH1L, SEPTIN1, and WASL).

The significant enrichment of the "male meiosis I", "female meiosis I", and "male gonad development" GO terms indicated that most of the changes observed in gonads at T2, rather than being merely linked with cell proliferation, were specifically due to the development of new gamete primordia, consistent with previous observations [12]. The gametogenic process is regulated by tight cell cycle checkpoints [46], which usually determine meiotic arrests at two distinct time points, i.e., prophase I [47] and metaphase II [48]. The anaphase-promoting complex/cyclosome (APC/C) is the key molecular player in these dynamics, controlling the ubiquitination and proteasome-mediated degradation of target cell cycle regulators such as cyclins and several chromatin-associated proteins. We observed the simultaneous up-regulation of DEGs that act both as promoters and inhibitors of meiotic progression at different stages, which may be due to the asynchronous status of developing germ cells in *Congeria* gonads. The specific functions of the most significant DEGs involved in the meiotic process are detailed in Table 2. Among these, we observed structural components of APC/C (i.e., ANAPC2, ANAPC5, ANAPC7, and CDC23), its key positive and negative regulators CDC20 and BUB1B, plus a number of genes involved in targeted protein turnover during meiosis (e.g., KLHL10, PSMD3, and UBB) [49]. The up-regulation of HSF2BP, MEIOB, MND1, and SYCP1, which are all required for efficient homologous recombination and DNA double strand break repair, and ASPM, which regulates meiotic spindle formation and cytokinesis, were also consistent with ongoing meiosis. The contrasting up-regulation of PKMYT1, which phosphorylates and inactivates CDK1, a key promoter of cell cycle progression, and of its upstream negative regulator PLK1, a direct activator of APC/C [50], also supported the presence of asynchronous meiotic divisions in *C. kusceri* gonads at T2. Similarly, cyclin A1, a meiosis-specific factor that prevents chromosome segregation and inhibits anaphase entry [51], was one of the DEGs displaying the highest levels of induction (FC = $60.61\times$).

**Table 2.** List of the most representative DEGs identified in gonads at T2 involved in the regulation of cell cycle progression.

| Gene Symbol | Full Name | FC | FDR *p*-Value | Role in Cell Cycle Progression and Meiosis |
|---|---|---|---|---|
| ANAPC2 | Anaphase Promoting Complex Subunit 2 | 8.49 | 0.0000206 | Component of the APC/C complex; promotes metaphase-anaphase transition by ubiquitinating and promoting proteasomal degradation of several cell cycle regulators. |
| ANAPC5 | Anaphase Promoting Complex Subunit 5 | 11.88 | 0.05 | Component of the APC/C complex; promotes metaphase-anaphase transition by ubiquitinating and promoting proteasomal degradation of several cell cycle regulators. |
| ANAPC7 | Anaphase Promoting Complex Subunit 7 | 5.48 | 0.0005 | Component of the APC/C complex; promotes metaphase-anaphase transition by ubiquitinating and promoting proteasomal degradation of several cell cycle regulators. |
| ASPM | Assembly Factor For Spindle Microtubules | 5.43 | 0.00248 | Regulates spindle assembly and meiotic progression. |
| AURKA | Aurora Kinase A | 2.13 | 0.00879 | Key regulator of various mitotic events. |
| BUB1B | BUB1 Mitotic Checkpoint Serine/Threonine Kinase B | 14.1 | 0.02 | Inhibits the APC/C complex, delaying the onset of anaphase and ensuring proper chromosome segregation. |
| CCNA2 | Cyclin A2 | 2.38 | 0.0000239 | Controls both the G1/S and the G2/M transition phases of the cell cycle. |
| CDC20 | Cell Division Cycle 20 | 20.39 | 0 | Co-activator of the APC/C complex. |
| CDC23 | Cell Division Cycle 23 | 23.98 | 0.03 | Component of the APC/C complex; promotes metaphase-anaphase transition by ubiquitinating and promoting proteasomal degradation of several cell cycle regulators. |
| HSF2BP | Heat Shock Transcription Factor 2 Binding Protein | 4.96 | 0.00293 | Modulates the localization of recombinases to meiotic double-strand break sites. |
| MEIOB | Meiosis Specific With OB-Fold | 4.08 | 0.03 | Stabilizes recombinases, allowing homologous recombination in meiosis I. |
| MELK | Maternal Embryonic Leucine Zipper Kinase | 5.28 | 0.00702 | Mediates the phosphorylation of CDC25B, promoting its localization to the centrosome and spindle poles. |
| MND1 | Meiotic nuclear division protein 1 homolog | 5.49 | $1.64 \times 10^{-9}$ | Stimulates homologous strand assimilation, which is required for the resolution of meiotic double-strand breaks. |
| MTBP | MDM2 Binding Protein | 13.1 | 0.000457 | Promotes the degradation of CDC25C and delays cell cycle progression through the G2/M phase. |
| PKMYT1 | Protein Kinase, Membrane Associated Tyrosine/Threonine | 4.26 | 0.0000116 | Negatively regulates the G2/M transition by phosphorylating and inactivating CDK1. |
| PLK1 | Polo Like Kinase 1 | 3.09 | 0.05 | Regulates centrosome maturation and spindle assembly, the removal of cohesins from chromosome arms, the inactivation of APC/C inhibitors, mitotic exit, and cytokinesis. |
| RACK1 | Receptor For Activated C Kinase 1 | 12.07 | 0 | Prolongs the G0/G1 phase of the cell cycle. |
| SYCP1 | Synaptonemal Complex Protein 1 | 3.62 | $3.89 \times 10^{-10}$ | Required for the formation of synaptonemal complexes during meiosis. |
| UBB | ubiquitin B | 15.64 | 0 | Controls the turnover of chromosomal proteins, regulating chromosome axis length during meiotic chromosome reorganization. |
| ZC3HC1 | Zinc Finger C3HC-Type Containing 1 | 3.35 | 0.00513 | Controls mitotic entry by mediating the ubiquitination and degradation of cyclin B1 |

Several DEGs were specifically associated with the production of sperm-associated axonemal proteins, such as dyneins, kinesins, tektins, and outer dense fiber proteins, as suggested by the enrichment of GO terms such as "sperm flagellum" and "spermatogenesis". A few upregulated tubulin genes were exclusively expressed in the gonads, suggesting they might be male-gonad-specific, as previously reported in other invertebrates [52]. One of the top upregulated DEGs was cyclin O (FC = 16.98×), which is specifically required for the generation of centrioles associated with motile cilia in multiciliated cells [53]. Several other orthologs of human testis-specific genes, often encoding structural components of spermatozoa, were overexpressed at T2, including SPATC1L, TSSK3, TSKK5, TEX10, TEX11, TEX33, TEX45, TEPP, and THEG. Another important class of upregulated DEGs at T2 were Piwi-like genes, which promote the function of spermatozoa by targeting, through piRNA-mediated interactions, a large number of spermatogenic genes [54]. Although these observations clearly supported the presence of enhanced spermiogenesis at T2, we need to note that the sex of the animals was not determined at the time of sampling. Consequently, the differential expression of the aforementioned testis-specific genes may also be linked, to some extent, to a different male-to-female ratio between the individuals sampled at the two time points.

Not surprisingly, the need for an increased production of cytoskeletal proteins to support meiosis determined a significant enhancement of gonad transcriptional activity at T2, perhaps triggered by the strong up-regulation of cyclin K (FC = 10.53×), an important regulator of the activity of RNA polymerase II [55]. We observed a high number of upregulated DEGs involved in pre-mRNA splicing, polyadenylation, and nuclear exportation (as evidenced by the significant enrichment of the GO terms "mRNA splicing, via spliceosome", "mRNA transport", and "mRNA polyadenylation"). This resulted, in turn, in an increased expression of genes involved in ribosome recruitment and translation, including eukaryotic translation initiation and elongation factors. The high abundance of newly synthesized tubulins most likely led to the overexpression of all the subunits of the TRiC complex (TCP1, CCT2, CCT3, CCT4, CCT5, CCT6, CCT7, and CCT8), even though the high activity of this chaperonin complex may also be interpreted as a means to allow the proper folding of CDC20, leading to APC/C activation [56].

The energetic requirements to support ongoing DNA replication and cell division and the intensified transcription and translation of mRNAs encoding cytoskeletal proteins had a profound effect on the metabolic rates of *C. kusceri* gonads. In fact, nearly all the genes encoding the key enzymes involved in both glycolysis (Figure 5A) and the tricarboxylic acid cycle (Figure 5B) were upregulated at T2. Although the magnitude of such alterations varied from gene to gene, the expression changes of some key regulators, such as phosphofructokinase [51], approached or even exceeded 10 folds. Crucially, some members of the pyruvate dehydrogenase complex (not shown in Figure 5), which connects glycolysis to the TCA cycle, were also significantly upregulated. Namely, these included the genes encoding the alpha and beta subunits of the E1 component (FC = 5.73× and 3.57×, respectively), the E2 dihydrolipoamide acetyltransferase component (FC = 9.91×), and the E3 lipoamide dehydrogenase component (FC = 8.83×), plus the regulatory component X (FC = 14.83×) and the regulatory subunit of the pyruvate dehydrogenase phosphatase (FC = 11.61×). Interestingly, pyruvate carboxylase, a key enzyme for both gluconeogenesis and lipogenesis, was also significantly upregulated at T2 (FC = 6.41×).

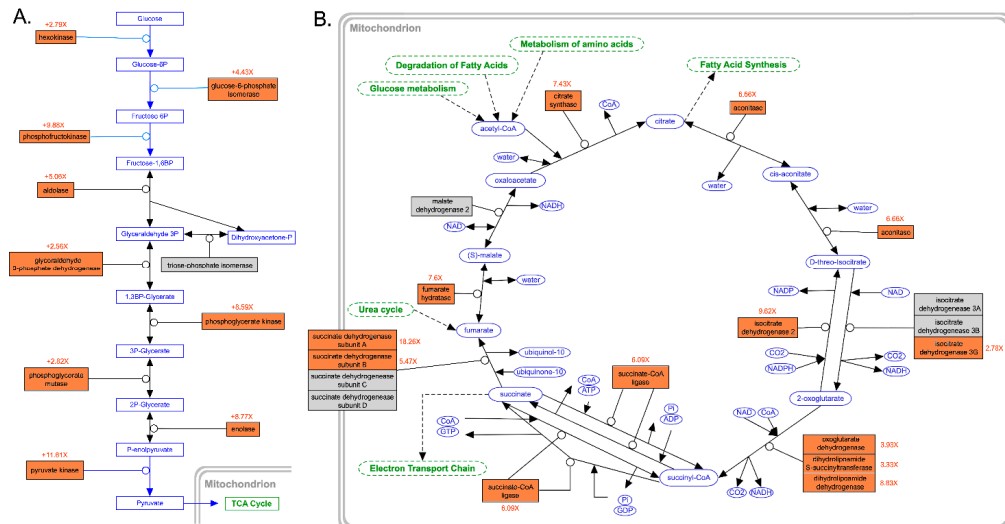

**Figure 5.** (**A**): Schematic representation of the main enzymatic reactions of glycolysis, modified from WikiPathways (https://www.wikipathways.org/instance/WP534, accessed 1 November 2022), summarizing the gene expression changes observed in this study in the comparison between T2 and T1. (**B**): Schematic representation of the main enzymatic reactions of the tricarboxylic acid cycle, modified from WikiPathways (https://www.wikipathways.org/instance/WP78, accessed 1 November 2022), summarizing the gene expression changes observed in this study in the comparison between T2 and T1. Differentially expressed genes that were upregulated at T2 are shown with a red background, whereas those that were not regulated are shown with a gray background. Fold change values, displayed in red, indicate the ratio between the expression levels observed at T2 and T1. Connected biological pathways are shown in green, molecules participating to reactions, either as cofactors, substrates or end products of enzymatic reactions, are shown in blue.

### 3.6. In-Depth Analysis of Differential Gene Expression in the Mantle

Although the mantle displayed a high number of DEGs, the overwhelming majority of tissue-specific responses were observed at T1 (Figure 2F), where the most outstanding changes were linked with shell biomineralization, consistent with the timing of shell deposition previously reported in *C. kusceri* [11]. This is in line with the hypothesis that the transition between the colder months of the winter and the summer season is the most favorable one for body growth, with a consequently important energetic investment in shell biomineralization.

We detected a high number of DEGs encoding shell matrix proteins (SMPs), often inferred to be directly involved in shell biomineralization due to their up-regulation in response to shell damage [57–63]. The most relevant classes of DEGs showing significant sequence homology with SMPs previously reported in three other bivalve species (*Pecten maximus*, *Crassostrea gigas*, and *Mytilus edulis*) are summarized in Figure 6A. These genes generally displayed significant over-expression at T1 compared with T2, often exceeding 5 or even 10 folds, which was fully consistent with the arrest of the shell growth process that usually occurs at the end of the summer.

Chitin is a major non-protein component of bivalve shells, and both chitin synthase and chitinase enzymatic activities have been previously implicated in shell formation and repair following experimental notching [64–67], which is also supported by the malformations observed during early development following exposure to chitinase inhibitors [68]. The importance of chitin in the context of *C. kusceri* shell deposition was confirmed by the up-regulation of four genes encoding chitinases, which only differed from each other due to the presence/absence of chitin-binding domains in their C-terminal ends. In total, 16 DEGs encoded proteins containing one or more chitin-binding domains, which generally had largely variable architectures (Figure 6B). While many of these were uncharacterized, others displayed significant homology with chitinases [68] and Pif proteins [63]. In three DEGs,

the chitin-binding domain was associated with a Von Willebrand factor A (vWA) domain, thereby showing the same domain combination found in *Mytilus* BMSP, a major component of mussel shells, able to bind the calcium carbonate crystals of aragonite and calcite in the nacreous layer [69].

Seven DEGs encode tyrosinases, which belong to a gene family that underwent significant expansion in the bivalve lineage [70]. In invertebrate animals, tyrosinases play important roles in a multitude of biological processes, including immune response (acting as prophenoloxidases) and hexoskeleton production, but they are also involved in bivalve shell biomineralization [71,72], for example by controlling the size of calcium carbonate crystals [73]. Some studies also suggest that the tyrosinases expressed in the mantle edge might contribute to melanin production, determining shell color [74]. However, since the shell of *C. kusceri* is not pigmented, as one of its many adaptations to the subterranean environment, we might expect such differentially expressed tyrosinase genes not to be involved in shell pigmentation but rather in the biomineralization process.

Five additional DEGs upregulated at T1 belonged to the amine oxidase gene family, which has been implicated in shell production during larval growth in *Pinctada fucata* [75]. One of the most significantly upregulated genes at T1 was beta-hexosaminidase (FC = 13.45×), whose enzymatic activity is involved in restructuring carbohydrate glycoside bonds during shell repair [76]. Two DEGs encode proteins with C1q and C-type lectin domains, respectively, which are usually associated with lectin-like molecules in innate immune recognition but are also found in important SMPs, such as the mussel major extrapallial fluid protein [77] and abalone perlucin [78].

Several protease inhibitors belonging to five distinct families, i.e., kazal-type, kunitz-type, tissue inhibitor of metalloprotease (TIMP), whey acidic protein (WAP), and alpha-2-macroglobulins (A2M), were upregulated at T1. As previously noted by other authors [79], protease inhibitors are a recurrent theme in shell biomineralization proteomes, where they are believed to prevent the premature degradation of extracellular matrix proteins by the numerous proteolytic enzymes present in the extracellular space. Among these five protease inhibitor families, the most represented one among the DEGs detected at T1, with 11 out of 12 upregulated members, was undoubtedly A2M. Although the precise role of A2M in the context of bivalve shell biomineralization has not been clarified yet, several orthologous sequences have been previously identified as SMPs, not just in bivalves but also in other molluscan classes [80].

Different peroxidases and peroxiredoxins were also differentially expressed, even though the former ones generally displayed higher fold change values. Although peroxidases, like tyrosinases, have been implicated in shell pigmentation, they may also play a dual role in bivalve biomineral-hydrogel formation [61]. Similarly, some studies strongly suggested that transgelin/calponin-like proteins, which were also found with three upregulated members at T1 in this study, are involved in the formation of aragonite crystals in the prismatic layer of the shell [81].

Due to the low primary sequence complexity displayed by the SMPs involved in determining shell architecture, collectively known as repetitive low complexity domain (RLCD) proteins, it is likely that several other uncharacterized DEGs displaying marked tissue specificity missed the minimum e-value threshold required for annotation.

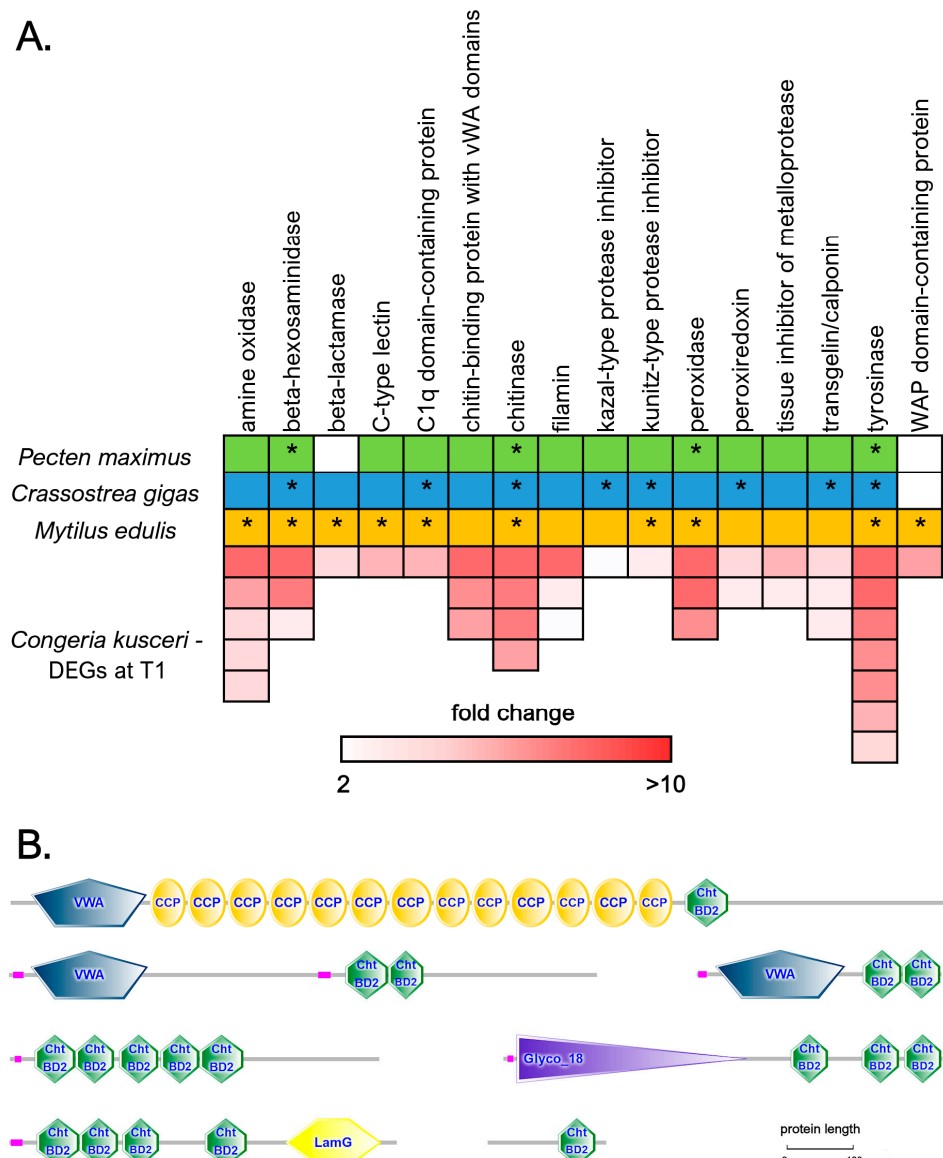

**Figure 6.** (**A**): A selected panel of differentially expressed genes in the *C. kusceri* mantle at T1, belonging to the same gene families previously described as mantle-specific in three other bivalve species (*Pecten maximus*, *Crassostrea gigas*, and *Mytilus edulis*), as reported by Yarra and colleagues [76]. Boxes colored green, blue, and orange indicate the identification of these genes as shell matrix proteins in any of the three species. Asterisks mark genes that are upregulated in response to shell damage and are thereby likely to play a key role in shell biomineralization. For *C. kusceri*, the colored boxes indicate the number of DEGs belonging to each gene family, and the color scale indicates the fold change value at T1. (**B**): A few examples of the structural architecture of differentially expressed chitin-binding proteins identified in the mantle of *C. kusceri* at T1. Cht BD2: chitin-binding domain; CCP: complement control protein domain; VWA: Von Willebrand factor type A domain; Glyco_18: chitinase II domain; LamG: laminin G domain. Low-complexity regions are marked in purple.

### 3.7. In-Depth Analysis of Differential Gene Expression in the Adductor Muscle

Compared with the three tissues examined above, the adductor muscle of *C. kusceri* displayed a limited transcriptional response (Figure 2D), and only a small number of the DEGs identified both at T1 and at T2 were tissue-specific (Figure 2F). Although the functional interpretation of the different expression profiles observed at the two time points was hampered by the low fraction of DEGs displaying meaningful annotations, both time points were characterized by the modulation of structural components of mus-

cle fibers. Most notably, an actin homologous to an adductor muscle-specific actin from *Placopecten magellanicus* was among the most strongly upregulated DEGs at T1, being over-expressed by nearly 100 folds and exceeding 2000 TPM [82]. This was accompanied by the up-regulation of other tissue-specific genes expressed at lower levels, including myosin light chain and collagen. At T2, numerous tissue-specific DEGs showing similarity with kielin/chordin proteins were identified. These often displayed high FC values, sometimes exceeding 1000 folds. While the role of the encoded proteins was uncertain, kielin/chordin-like proteins are generally thought to be involved in extracellular matrix organization by modulating the activity of BMP (Bone Morphogenetic Protein) signaling [83]. Other DEGs suggestive of ongoing tissue remodeling in *C. kusceri* adductor muscle at T2 were peroxidasin, involved in strengthening muscle fibers [84], twitchin, a giant muscle fiber protein previously reported in mollusks [85], and iduronate 2-sulfatase, an heparan sulfate modifying enzyme. The latter gene could be particularly relevant, considering the key role played by heparan sulfate in regulating muscle motor activity [86]. Notably, some genes encoding short-chain collagen-like proteins and enzymes involved in collagen biosynthesis (e.g., prolyl 4-hydroxylase) were also upregulated at T2. At the same time, some tyrosinase-like proteins, not overlapping with those previously reported in the mantle and displaying strong tissue specificity, were also strongly overexpressed. Despite some limitations in our current understanding of the role played by these DEGs in regulating adductor muscle physiology, all the information reported above was consistent with a significant modification of *Congeria*'s muscle architecture and function throughout the summer.

Considering the fundamental function played by the adductor muscle in regulating valve opening, we speculate that such changes may be linked with alterations in filter-filtering behavior and consequently with the need for improved or diminished adductor muscle activity, perhaps in response to food availability and/or variable water levels and temperature.

### 3.8. In-Depth Analysis of Differential Gene Expression in the Digestive Gland

The digestive gland was, by far, the tissue displaying the least significant fluctuations in gene expression levels, with little evidence of functional enrichment, both within the set of DEGs upregulated at T1 and T2. In early June, more than half of DEGs displayed a FC value comprised between 2 and 3, denoting a much smaller magnitude of regulation compared with other tissues (Figure 2E). Nevertheless, compared with the other tissues, a high fraction of DEGs were strongly tissue-specific (Figure 2F). Among these, the most noteworthy were those upregulated at T2. Indeed, we noticed a strong up-regulation of several digestive enzymes, which in some cases exceeded 100-fold activation. These included different chymotrypsins and a single carboxypeptidase, two classes of enzymes that have been previously implicated in digestive processes in bivalve mollusks [87,88]. Moreover, several DEGs encode proteins carrying multiple chitin-binding domains at T2. Although nutrient availability was not monitored in the weeks prior to sampling, these remarkable changes may, to some extent, depend on the increased availability of chitin-rich food sources at T2, which may in turn be due to the variable water influx from the surface. Alternatively, the increased production of chitin-binding proteins may be linked with the production of chitin-rich mucus with a protective role, as previously evidenced in ascidians [89].

Although we had previously evidenced that the digestive gland was involved in the bioaccumulation of five Picorna-like viruses throughout the summer [18], we did not observe a significant enrichment of genes involved in immune response, except for a single MPEG-like gene. The perforin-like protein encoded by this gene may act in a similar fashion to the components of the terminal pathway of the complement system [90], eliminating infected cells [91].

## 4. Conclusions

In this study, we provide, for the first time, a detailed overview of the transcriptional changes that characterize a key period in the life cycle of *C. kusceri*. In fact, the study period covers the transition from warmer summer months, where shell growth reaches its peak and gamete maturation is completed prior to intense release, to autumn, when the decrease in water temperature and other concomitant alterations lead to the restart of the gametogenic cycle. Our findings, which highlighted a remarkable switch in the use of the energy budget from tissue growth at T1 to the restoration of spent gonads at T2, shed new light on the molecular mechanisms underpinning the annual life cycles of this species.

Due to the endangered status of *C. kusceri*, our experimental design could only include five adult individuals per sampling, whose sex was not determined. Hence, despite the presence of high statistical significance, we cannot rule out the possibility that a fraction of the DEGs characterized by small FC values were due to: (i) inter-individual variability of expression linked with within-population genetic or epigenetic variation; (ii) a different male:female ratio in the individuals sampled at the two time points; or (ii) gene presence/absence variation (PAV). Although still unreported in bivalves other than *Mytilus* spp., the gene PAV is likely common [37,92] and might be relevant also in epigean members of Dreissenidae, which display high heterozygosity rates [39]. Moreover, this study was focused on the investigation of transcriptomic responses in adult individuals. Therefore, our observations may not necessarily apply to juveniles, which may be more susceptible than adults to seasonal variations. This will be an important point to be addressed by future studies since a successful settlement of juveniles and their ability to reach sexual maturity are key aspects to supporting the long-term survival of this species.

Despite suffering from the aforementioned limitations, this study provides important information that complements previously collected morphological and physiological observations, improving our knowledge of the biology of this highly endangered bivalve species. In detail, the results of this study were fully consistent with previous investigations on the physiology, life cycle, and reproductive strategy (related to K selection) of the species [11,12,20], but revealed the molecular determinants that most likely underpin observed morphological and physiological changes. Due to ongoing habitat degradation and rapid population decline, conservation plans are urgently needed to preserve *C. kusceri* in its few known type locations. With this respect, the establishment of molecular resources such as the annotated transcriptome and associated tissue expression atlas reported in this study might represent a valuable tool to support improved biomonitoring campaigns as well as to plan further studies aimed at clarifying specific aspects of *Congeria* biology that are still poorly understood. Similar approaches have been successfully applied to similar models, i.e., endangered epigean freshwater unionid mussels, which, despite being phylogenetically distant from *Congeria* (i.e., they belong to the subclass Palaeoheterodonta), are exposed to similar threats and are largely represented in the IUCN Red List of Threatened Species [93]. For example, molecular tools have already been successfully used to investigate the genetic structure of remnant populations [94], helping to define conservation units [95], or to develop environmental DNA-based tools for non-invasive biomonitoring [96], following an approach that has also recently been proposed for *C. jalzici* [97]. In general, due to their broad applicability and rapidly decreasing costs, next-generation sequencing technologies have been identified as one of the most critical factors in improving the quality of studies on the conservation of freshwater bivalves [98]. Based on these premises, we believe that the conservation status of *C. kusceri* might significantly benefit from the availability of omics resources in the near future.

**Supplementary Materials:** The following supporting information can be downloaded at: https://www.mdpi.com/article/10.3390/d15060707/s1. Table S1: number of clean reads used for the de novo assembly of the *Congeria kusceri* transcriptome as well as for subsequent gene expression analyses; Table S2: complete functional annotation of assembled *Congeria kusceri* contigs, including the best UniProt hit and associated description, Gene Ontology terms, Pfam domains, orthoDB

assignment and description, and Reactome pathway; Table S3: significantly enriched Gene Ontology biological process terms; Table S4: significantly enriched Gene Ontology cellular component terms; Table S5: significantly enriched Gene Ontology molecular function terms; Figure S1. Summary of the most significantly enriched Gene Ontology cellular component terms associated with differentially expressed genes. Only the terms achieving a FDR-corrected enrichment *p*-value lower than $1 \times 10^{-8}$ in at least one pairwise comparison, paired with a difference between the observed and expected number of observed DEGs associated with each GO term >5, are shown. The complete list of all significantly enriched Gene Ontology cellular component terms is reported in Table S4. DG: digestive gland; MUS: adductor muscle; GON: gonads; GIL: gills; MAN: mantle; Figure S2. Summary of the most significantly enriched Gene Ontology molecular function terms associated with differentially expressed genes. Only the terms achieving a FDR-corrected enrichment *p*-value lower than $1 \times 10^{-8}$ in at least one pairwise comparison, paired with a difference between the observed and expected number of observed DEGs associated with each GO term >5, are shown. The complete list of all significantly enriched Gene Ontology molecular function terms is reported in Table S5. DG: digestive gland; MUS: adductor muscle; GON: gonads; GIL: gills; MAN: mantle.

**Author Contributions:** Conceptualization, M.G., T.R. and S.P.; methodology, A.S. and S.G.; software, S.G.; validation, A.S.; formal analysis, A.S., S.G. and C.M.; investigation, A.S. and C.M.; resources, S.P.; data curation, A.S.; writing—original draft preparation, M.G. and A.S.; writing—review and editing, all authors; visualization, M.G.; supervision, M.G. and A.P.; project administration, M.G. and A.P.; funding acquisition, M.G. All authors have read and agreed to the published version of the manuscript.

**Funding:** This work was supported by the Microgrants funding program of the University of Trieste—a transcriptomic investigation of the endemic cave-dwelling "living fossil" bivalve *Congeria kusceri*.

**Institutional Review Board Statement:** Not applicable.

**Data Availability Statement:** Raw sequencing data have been deposited in the NCBI Sequence Read Archive (SRA) under the BioProject ID PRJNA704250 (SRR13767952-SRR13767961).

**Acknowledgments:** The authors thank the Croatian Ministry of Economy and Sustainable Development for permission to collect specimens of *C. kusceri* for the purposes of this study (UP/I-612-07/18-48/81; 517-07-1-1-18-4).

**Conflicts of Interest:** The authors declare no conflict of interest. The funders had no role in the design of the study, in the collection, analysis, or interpretation of data, in the writing of the manuscript, or in the decision to publish the results.

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
