# Peer review of "Variation of Gene Expression in the Endemic Dinaric Karst Cave-Dwelling Bivalve Mollusk Congeria kusceri during the Summer Season"

_diversity, doi:10.3390/d15060707_

Round 1

Reviewer 1 Report

Why authors chose the samples between early June and late September? The title might be better to more specific on over the course of the summer season.

As authors stated the alkalinity displays sudden changes during summer seasons, it is important to measure the alkalinity. Did author measure the alkalinity on the sampling place at different time?

Can authors describe the potential effects of alkalinity on the bivalve species?

I am not understand why authors use five individual as pooled sample instead of doing replicates, because the RNA-seq analysis needs at least three replicates to improve the accuracy of the results.

I think it is necessary to validate the RNA-seq data by qPCR analysis, please add the qPCR results.

Line115: According to water level measurements conducted as part of this study, a drop in water level of approximately 42 cm was observed at T2, compared with T1. What’s the difference of water chemistry parameters between T2 and T1?

L232The figure B and C do not match the text description.

L652Supplementary information website link cannot be opened.

L145-146: Why all the contigs that did not reach the arbitrary threshold of 5 Transcript Per Million (TPM) were removed.

L237: how to determine that out of the 10,101 DEGs in this study, 3,104 (~30.7%) ware shared by more than one tissue.

L248-249: At which period was the tissue specific DEGs measured in each section of the ms?

Tables: I suggest authors to add table head and end line, and add the name of each column in Table 1.

How authors chose the genes shown in Figure 5?

Figure 4 is not clear, please provide high resolution picture.

Moderate editing of English language

Author Response

Why authors chose the samples between early June and late September? The title might be better to more specific on over the course of the summer season.

The rationale for this choice is explained in detail in section 3.2 and further detailed in Figure 2 (now renumbered as Figure 3). We selected two time periods characterized by significantly different phases of the reproductive cycle and shell biomineralization, which recapitulate the key points of the annual growth cycle of C. kusceri. We recognize that the use of the term “seasonal” in the title may be misleading, as we did not perform a time course experiment covering 12 months, and modified it accordingly. The updated title now reads, also based on an additional suggestion provided by reviewer #2, “Variation of gene expression in the endemic Dinaric karst cave-dwelling bivalve mollusk Congeria kusceri during the summer season”

As authors stated the alkalinity displays sudden changes during summer seasons, it is important to measure the alkalinity. Did author measure the alkalinity on the sampling place at different time?

Although this parameter was not monitored during the sampling season, variations of alkalinity show remarkable seasonal variations and are therefore  predictable at the Jama u Predolcu, as previously demonstrated by Puljas et al. 2014 (Malacologia 57(2):353-364). Indeed, water alkalinity reaches its minimum during the summer season (i.e. approximately starting from May), reaching values as low as 150/160 mg CaCO3/L, which remain fairly stable until September. In October, in parallel with the decrease in water temperature, these values increase, usually exceed 230/240 mg CaCO3/L and remain above 200 mg CaCO3/L until the following May. In the study mentioned above, the differences in water temperature (t = 4.929, p < 0.001) and alkalinity (t = 3.702, p = 0.004) were highly significant in the six months when the animals were not growing and breeding and in the six months when they were growing and breeding. Both our sampling time points were within the period where water alkalinity was expected to be at its minimum, well below 200 mg CaCO3/L.

Can authors describe the potential effects of alkalinity on the bivalve species?

In general, bivalve shells are composed by more than 95% CaCO3 and by a small fraction of organic matrix. Clearly, water alkalinity (which depends on CaCO3 concentration) is expected to have profound effects on different aspects of shell biomineralization, including CaCO3 crystals nucleation, their growth and inhibition. However, shell formation is a complex process, which is guided by the activity of specialized mantle cells, which are able to synthesize amorphous calcium carbonate (presumed to be a precursor of shell minerals), and a number of shell matrix proteins, which can act both as positive or as negative regulators of this process. Therefore, besides alkalinity, shell biomineralization is critically also regulated by water temperature.

As discussed in a previous publication (Puljas et al. 2014), temperature appears to be the main trigger of shell growth in C. kusceri, mirroring the patterns observed in the epigean related species Dreissena polymorpha (Karatayev et al., 2006) and Mytilopsis leucophaeata (Verween et al., 2010). Although the molecular aspects of shell biomineralization in C. kusceri are still not entirely clear, CaCO3 uptake reached its maximum in periods of lower theoretical bioavailability in the water. However, this observation requires physiological elaboration as well as investigation of other factors that may be responsible for such a decrease in alkalinity.

We modified the text and improved the discussion of this specific point in the main text.

I am not understand why authors use five individual as pooled sample instead of doing replicates, because the RNA-seq analysis needs at least three replicates to improve the accuracy of the results.

In principle, we would agree with the strategy proposed by the reviewer, as the use of biological replicates improves statistical power and allows the detection (and removal) of outlier samples. However, body size was a limiting factor in this study since, as mentioned in the materials and methods section, sampled adult individuals ranged in shell size from 9.6 to 12 mm. Hence, some of the soft tissues sampled (the adductor muscle and gonads in particular) were close to 1 mm in size, preventing to obtain sufficient amounts of total RNA to allow the preparation of libraries of good quality for all individuals. We also anticipated the need to carry out additional cleanup steps to obtain high quality RNA, based on our extensive experience on RNA extraction from bivalve tissues, and planned a pooling strategy beforehand to overcome potential sample loss issues during RNA purification.

Therefore, albeit being certainly not optimal, our strategy was a compromise between the impossibility of obtaining high quality libraries from single individuals and the need to mitigate inter-individual differences, which we tried to achieve by using a number of individuals as high as possible (and allowed by authorities). Indeed, while we could have hypothetically obtained 3 or 5 biological replicates from additional batches of C. kusceri individuals, we need to keep in mind that this is a critically endangered, red-listed species. Unfortunately, RNA-seq is by its own nature an invasive method, which requires the sacrifice of the sampled animals. We did not feel that sacrificing a higher number of individuals was reasonable in this case.

Please note that we made sure to use a statistical test (the Kal’s Z-test) specifically developed to run DGE analysis on samples lacking biological replicates and used stringent criteria for the selection of DEGs (i.e. FDR-corrected p-values < 0.05, paired with FC values > 2 or < -2) with the aim to exclude more subtle alterations that could be linked with the presence of outlier samples in the pool.

I think it is necessary to validate the RNA-seq data by qPCR analysis, please add the qPCR results.

We respectfully disagree with this requirement. Although RNA-seq and qPCR have a different dynamic range and are based on a different quantification method, multiple studies have demonstrated that the results produced by the two methods are highly correlated. Although qPCR validation may represent an added value in some cases, we don’t think that validating RNA-seq using the very same RNA samples by qPCR is one of these.

The current scientific consensus, summarized by the editorial recently published by Tom Coenye (https://doi.org/10.1016%2Fj.bioflm.2021.100043) reads: “the data available suggest that RNA-seq methods and data analysis approaches are robust enough to not always require validation by qPCR and/or other approaches, although there are situations where this may be of added value”.

Ideally, validation by qPCR would be extremely valuable only when performed on a different batch of samples, i.e. in our case on additional individuals, or in individuals sampled in a different year. However, due to the aforementioned limitations (i.e. primarily due to the critically endangered status of the target species), no additional individuals could be sampled.

Line115: According to water level measurements conducted as part of this study, a drop in water level of approximately 42 cm was observed at T2, compared with T1. What’s the difference of water chemistry parameters between T2 and T1?

As mentioned above, water chemistry parameters were not specifically analyzed in the frame of this study. The only information available is as follows: the average water temperature on sampling day T1 was 16.7°C, while the dissolved oxygen concentration was 8.6 mg/L. On sampling day T2, water temperature was 17.2°C and dissolved oxygen was 6.6 mg/L. Both parameters were measured using a portable multimeter (Mettler Toledo X-matePro MX300).

Historical long-term data are scarce for this site, but a few information available for the Jama u Predolcu derives from a previous publication by Jovanovic Glavas and colleagues, who monitored this location for two consecutive years (International Journal of Speleology 46 (1) 13-22, 2017), recording water levels and temperatures. Besides the previous year-round recording of water temperature and alkalinity reported in Puljas et al. 2004, a comprehensive analysis of several water chemistry parameters for this site is reported  for 2009 in Rada and Rada (talian Journal of Zoology, March 2012; 79(1): 105–110). We added this reference to the main text to provide the readers with additional context.

L232:The figure B and C do not match the text description.

Thank you for pointing this out, this was corrected.

L652:Supplementary information website link cannot be opened.

We believe that the reviewer is referring to the link www.mdpi.com/xxx/s1. This is a placeholder that is included in the template of all MDPI manuscripts and will be updated by the editorial team once the manuscript will be accepted for publication and posted online. The supplementary materials should be available to the reviewer through the main reviewer portal of the MDPI website.

L145-146: Why all the contigs that did not reach the arbitrary threshold of 5 Transcript Per Million (TPM) were removed.

We are sorry for not having explained properly the reason for this choice. This is part of our standard protocol for transcriptome assembly cleanup. As it is usual in bivalves (and all filter-feeding organisms in general), exogenous contamination is always possible due to the presence of food particles, bacteria, algae, protozoans and other unicellular organisms present in the water column or associated with the shell or with tissues themselves. Therefore, total extracted RNA may include a small fraction of exogenous mRNAs not belonging to the target species. Nevertheless, these are expected to be poorly abundant and can be therefore safely removed by applying a cut-off of expression, i.e. by discarding all contigs represented by a very low number of reads. This process also leads to other benefits, allowing the removal of a high number of short contigs derived from endogenous RNAs, but with little or no biological meaning in the experimental context. These include fragmented portions of long mRNAs expressed at negligible levels and, above all, non-coding RNAs deriving from spurious intergenic transcription, which in our experience is a rather common phenomenon in bivalves. The threshold used for this removal is arbitrary, but was based on our previous experience with bivalve transcriptomes. To validate the reliability of this approach, we need to remark the fact that the BUSCO completeness score of the filtered transcriptome was 93.5%, i.e. extremely high and did not decrease significantly compared with the preliminary transcriptome assembly, which demonstrates that no meaningful coding transcripts were discarded during this process. We added a few sentences to the materials and methods section to briefly explain this choice and referenced a few of our studies where we used the same approach.

L237: how to determine that out of the 10,101 DEGs in this study, 3,104 (~30.7%) ware shared by more than one tissue.

This was simply determined by comparing the lists of DEGs from each individual sample using a Venn diagram approach. We originally considered the option to include Venn diagrams as an additional figure, but this was later dropped, as the different tissues displayed markedly different (and sometimes opposed) trends, and we would have needed to add two distinct Venn diagrams (one for the DEGs upregulated at T1, another one for those upregulated at T2). Nevertheless, those would have confused the reader, since 10,101 is the number of all DEGs regardless of their up-regulation at T1 or T2.

L248-249: At which period was the tissue specific DEGs measured in each section of the ms?

Tissue-specificity was separately evaluated at T1 and T2, i.e. T2 samples were excluded for the identification of tissue-specific DEGs at T1 and vice versa.

Tables: I suggest authors to add table head and end line, and add the name of each column in Table 1.

We did not add column names, since the name of each metric is self-explanatory and the values contained in the second column represent different types of measurements (e.g. nucleotides, number of contigs, etc.). Table 1 will be formatted according to journal requirements based on the indications provided by the editorial staff during proofreading.

How authors chose the genes shown in Figure 5?

As the figure caption explains, we represented the gene families previously described as mantle-specific in three other bivalve species (Pecten maximus, Crassostrea gigas and Mytilus edulis), as reported by Yarra and colleagues in their publication.

Figure 4 is not clear, please provide high resolution picture.

We had a similar request from another reviewer. We realize that this figure is packed with information, so its readability, while embedded in a pdf file, might be reduced. Nevertheless, we provided the figure in very high resolution (10000x5000 pixels), to make sure that this figure will be comfortably readable (with the possibility of zooming-in) in the online version of the manuscript.

Reviewer 2 Report

This paper is very interesting, authors demonstrated that gene expression was evaluated in the cave-dwelling bivalve mollusk Congeria kusceri during the seasonal variation process. Authors provide an overview of the modulation of gene expression recorded in five key tissues (mantle, gonads, gills, adductor muscle and digestive gland) by RNA-sequencing between early June and late September. They found the presence markedly tissue-specific responses, with the most dramatic changes affecting gonads, mantle and gills. Moreover, a drastic switch found identified in the use of energy budgets between the two periods, with evidence of ongoing shell growth and high metabolic activity in mantle and gills at the early time point, followed by a massive redirection of all available energy to the gonads for the generation of new gamete primordia in early autumn. 

However, it was still revise such as:

1.Title revised it to “Seasonal variation of gene expression in the cave-dwelling bivalve mollusk Congeria kusceri in Italy/Southern Europe.

2. Images require higher clarity.

3.Discussion should focus on the relationship between the phenomena and the genotypes.

Author Response

This paper is very interesting, authors demonstrated that gene expression was evaluated in the cave-dwelling bivalve mollusk Congeria kusceri during the seasonal variation process. Authors provide an overview of the modulation of gene expression recorded in five key tissues (mantle, gonads, gills, adductor muscle and digestive gland) by RNA-sequencing between early June and late September. They found the presence markedly tissue-specific responses, with the most dramatic changes affecting gonads, mantle and gills. Moreover, a drastic switch found identified in the use of energy budgets between the two periods, with evidence of ongoing shell growth and high metabolic activity in mantle and gills at the early time point, followed by a massive redirection of all available energy to the gonads for the generation of new gamete primordia in early autumn.

We thank the reviewer for his/her positive assessment of our manuscript.

However, it was still revise such as:

1.Title revised it to “Seasonal variation of gene expression in the cave-dwelling bivalve mollusk Congeria kusceri in Italy/Southern Europe”.

Thank you for this suggestion. The title was modified as follows based on this suggestion and another suggestion provided by reviewer #1: “Variation of gene expression in the endemic Dinaric karst cave-dwelling bivalve mollusk Congeria kusceri during the summer season”. We chose to indicate the Dinaric karst region, as the area of distribution of Congeria kusceri goes past country borders, covering regions located in Croatia and Bosnia-Herzegovina.

  1. Images require higher clarity.

We guess that this comment is linked with the resolution of the figures found embedded in the pdf version of this manuscript. Due to the high density of information, Figure 4 (now renumbered as Figure 5) might be particularly difficult to read. We provided all figures in high resolution, according to authors’ guidelines, which should ensure that they are fully readable (with the possibility to zoom-in) in the online version of the manuscript. We will work closely with the editorial team to make sure that the visualization of figures in the final version will be clear.

3.Discussion should focus on the relationship between the phenomena and the genotypes.

Thank you for this comment. We modified some points of the discussion and conclusion section according to this comment, further highlighting the fact that the extent by which Congeria may display individual responses linked with sex, age or genotype are unknown.

Reviewer 3 Report

The manuscript provides very interesting data for the transcriptome profile of a cave bivalve in relation to seasonality. It is generally well written, whereas the analyses are very well stated. Nevertheless, I do not agree with the structure of Results and Discussion together. I believe it would be better to be presented in separate sections. Further there are 2 major points that are not well discussed, as follows:

1. The authors discuss the timing of the two samplings in relation to chemico-physical fluctuations to explain the different trends of gene expression. However, they did not obtain any chemico-physical measurements. On the other hand the collection times presented very small temperature difference. How is gene expression difference explained in relation to temperature? The values that the authors discuss at lines 260-262 do not correspond to the temperature values reported in lines 111-113. This part has to be revised and further discussed.

2. The authors mention in the abstract and in the introduction that the studied species suffers from populations declines and therefore their study will enlighten the limited biological knowledge of the species that is expected to contribute towards the conservation. This statement is of course totally correct and true. However there is nothing in the discussion how their findings may support the conservation of the species. Thus I suggest to include in the discussion a part discussing the conservation in relation to the obtained data.

Also a few minor points

In introduction please explain the difference between stygobionts or as stygophiles

In Materials and Methods, since permits are referred, the permission code/number should be also provided

I believe Figure 2 should be separated and placed in Materials and Methods

Author Response

The manuscript provides very interesting data for the transcriptome profile of a cave bivalve in relation to seasonality. It is generally well written, whereas the analyses are very well stated. Nevertheless, I do not agree with the structure of Results and Discussion together. I believe it would be better to be presented in separate sections.

We thank the reviewer for his/her positive assessment of our manuscript. Based on the authors’ guidelines, which read, under the section linked with discussion, “This section may be combined with Results”, this journal allow the authors to choose whether to present the results and discussion as two distinct sections or to combine them in a single section. This has been a matter of debate among the authors during the initial drafting of the manuscript, and we all agreed that in this case, due to the need to separately discuss the results obtained for the different tissues, a combined results + discussion section would have been the best suited option for our manuscript. We still feel this was the most reasonable choice. Therefore, unless a division between the results and discussion sections is explicitly required by the editors, we would prefer to keep the manuscript structure in its current form.

Further there are 2 major points that are not well discussed, as follows:

  1. The authors discuss the timing of the two samplings in relation to chemico-physical fluctuations to explain the different trends of gene expression. However, they did not obtain any chemico-physical measurements. On the other hand the collection times presented very small temperature difference. How is gene expression difference explained in relation to temperature? The values that the authors discuss at lines 260-262 do not correspond to the temperature values reported in lines 111-113. This part has to be revised and further discussed.

The reviewer correctly pointed out that we did not obtain any chemico-physcal measurements at the time of sampling, besides water temperature. Nevertheless, according to published literature, and despite the presence of some fluctuations linked with rainfall, the variation of temperature and water alkalinity are fairly predictable at the Jama u Predolcu (see Puljas et al. 2014, Jovanovic Glavas et al. 2016, Rada and Rada 2011), so we can assume that the general trends of relevance for this study (i.e. high water temperature and low water alkalinity during the summer season) are well-established data that can be repeatedly observed each year. Indeed, cave habitats share some common characteristics: lack of light, relatively small amounts of food, all of which comes from outside the caves and is brought in by sinking rivers and percolating water, very high and stable relative humidity, which depends on air flow and circulation in the caves, relatively stable air temperature, which is usually the average annual temperature of the cave area but also depends on the morphology, hydrology, and proportions of the cave, and relatively stable water temperature. The temperature variation at this site of sampling (Jama u Predolcu ) is Δ = 7.75°C.

We expanded the text to provide additional information. 19°C represent the peak temperature recorded during the summer season, whereas the slightly lower temperatures recorded at the time of sampling were quite in line with the ranges reported in the same periods by previous studies (Puljas et al. 2014, Jovanovic Glavas et al. 2016, Rada and Rada 2011).

  1. The authors mention in the abstract and in the introduction that the studied species suffers from populations declines and therefore their study will enlighten the limited biological knowledge of the species that is expected to contribute towards the conservation. This statement is of course totally correct and true. However there is nothing in the discussion how their findings may support the conservation of the species. Thus I suggest to include in the discussion a part discussing the conservation in relation to the obtained data.

Thank you for this suggestion, that we feel was very important. We agree with the reviewer that this specific point would require some additional space in the discussion, so the text was expanded accordingly.

Also a few minor points

In introduction please explain the difference between stygobionts or as stygophiles

We added a brief sentence to provide a definition of the two terms. Ecologically, aquatic cave animals can be divided into stygobionts (species adapted to caves and restricted to underground waters), stygophiles (species that inhabit caves and complete their entire life cycle there, but also occur in similar habitats in open waters), and stygoxenes (species that are common in caves but must leave the cave to feed or reproduce).

In Materials and Methods, since permits are referred, the permission code/number should be also provided

I believe Figure 2 should be separated and placed in Materials and Methods

The permit number was already listed in the acknowledgements section “The authors thank the Croatian Ministry of Economy and Sustainable Development for permis-sion to collect specimens of C. kusceri for the purposes of this study (UP /I-612-07/18-48/81; 517-07-1-1-1-18-4).”, but based on the reviewers’ comment it was also added to the materials and methods section.

Figure 2 was split as per reviewer’s request, and one panel (the one showing the geographical location of sampling) was moved to the materials and methods section. All other figures were re-numbered accordingly.

Reviewer 4 Report

The manuscript titled "Seasonal variation of gene expression in the cave-dwelling bivalve mollusk Congeria kusceri" (Diversity- 2368252) aims the difference of gene expression, using RNA-sequencing, between early June and late September in different tissues of the bivalve mollusk Congeria kusceri from a Croatian cave.

In general, this work is a very interesting paper and it provides undoubtedly notable results that make a new contribution to better understand of how the environmental factors in warmer summer and autumn months and other concomitant alterations lead to the restart of the gametogenic cycle in this mollusk.

The paper is very well written and in my opinion, the results are solid and carefully interpreted and discussed.

Approach, methodologies and analysis used in this work are considered satisfactory. Language is clear and understandable. Figure are of good quality. Results are very clearly presented and the organization of the paper meets the criteria of the journal.

The conclusions are drawn appropriately based on the data presented.

I want to express my appreciation to the authors for their excellent paper.

I support its publication after minor revision.

Minor comments

In Table1, please change “avarege” with “average”

Table1 Please, add information to N50 and to Ex90N50. Are they “(nt)”?

Table 2 Please, change all commas with dots

Reference n. 60 L827 seems to be wrong. Please, correct with surname and dotted name of the authors.

Author Response

The manuscript titled "Seasonal variation of gene expression in the cave-dwelling bivalve mollusk Congeria kusceri" (Diversity- 2368252) aims the difference of gene expression, using RNA-sequencing, between early June and late September in different tissues of the bivalve mollusk Congeria kusceri from a Croatian cave.

In general, this work is a very interesting paper and it provides undoubtedly notable results that make a new contribution to better understand of how the environmental factors in warmer summer and autumn months and other concomitant alterations lead to the restart of the gametogenic cycle in this mollusk.

The paper is very well written and in my opinion, the results are solid and carefully interpreted and discussed.

Approach, methodologies and analysis used in this work are considered satisfactory. Language is clear and understandable. Figure are of good quality. Results are very clearly presented and the organization of the paper meets the criteria of the journal.

The conclusions are drawn appropriately based on the data presented.

I want to express my appreciation to the authors for their excellent paper.

I support its publication after minor revision.

We thank the reviewer for his/her positive assessment of our manuscript.

Minor comments

In Table1, please change “avarege” with “average”

Thank you for pointing out this typo. The table was corrected.

Table1 Please, add information to N50 and to Ex90N50. Are they “(nt)”?

Yes, these two metrics related to contig length, so they are (nt). We added this information to the table ad placed a short definition of N50 in the main text.

Table 2 Please, change all commas with dots

Thank you for highlighting this mistake, that we amended in the revised version.

Reference n. 60 L827 seems to be wrong. Please, correct with surname and dotted name of the authors.

The reviewer is correct, we believe this was a formatting error linked with the use of a reference manager software. We modified the names of the authors accordingly.

Round 2

Reviewer 3 Report

The authors addressed my comments, the manuscript may be published